

# Origins and characterization of CO and O$_3$ in the African upper troposphere.

Victor Lannuque[1,a], Bastien Sauvage[1], Brice Barret[1], Hannah Clark[4], Gilles Athier[1], Damien Boulanger[2], Jean-Pierre Cammas[5], Jean-Marc Cousin[1], Alain Fontaine[1], Eric Le Flochmoën[1], Philippe Nédélec[1], Hervé Petetin[6], Isabelle Pfaffenzeller[2], Susanne Rohs[3], Herman G.J. Smit[3], Pawel Wolff[1] and Valérie Thouret[1]

[1]Laboratoire d'Aérologie, CNRS, UPS, Toulouse, France
[2]Observatoire Midi-Pyrénées, Université de Toulouse, CNRS, UPS, Toulouse, France
[3]Forschungszentrum Jülich GmbH, Institut für Energie- und Klimaforschung, IEK-8 Troposphere, 52425 Jülich, Germany
[4]IAGOS-AISBL, 98 Rue du Trône, Brussels, 1050, Belgium
[5]Observatoire des Sciences de L'univers de la Réunion, UMS3365, la Réunion, France
[6]Barcelona Supercomputing Center, Barcelona, Spain
[a]Now at: CEREA, Joint Laboratory École des Ponts ParisTech/EDF R&D, Université Paris-Est, 77455 Marne-la-Vallée, France

*Correspondence to*: Victor Lannuque (victor.lannuque@enpc.fr) and Bastien Sauvage (bastien.sauvage@aero.obs-mip.fr)

**Abstract.**

Between December 2005 and 2013, the In-service Aircraft for a Global Observing System (IAGOS) program produced almost daily in situ measurements of CO and O$_3$ between Europe and southern Africa. IAGOS data combined with measurements from the IASI instrument onboard the Metop-A satellite (2008-2013) are used to characterize meridional

distributions and seasonality of CO and O$_3$ in the African upper troposphere (UT). The FLEXPART particle dispersion model and the SOFT-IO model which combines the FLEXPART model with CO emission inventories are used to explore the sources and origins of the observed transects of CO and O$_3$.

We focus our analysis on two main seasons: December to March (DJFM) and June to October (JJASO). These seasons have been defined according to the position of Intertropical Convergence Zone (ITCZ), determined using in situ measurements

from IAGOS. During both seasons, the UT CO meridional transects are characterized by maximum mixing ratios located 10° from the position of the ITCZ above the dry regions inside the hemisphere of the strongest Hadley cell (132 to 165 ppb at 0-5°N in DJFM and 128 to 149 ppb at 3-7°S in JJASO), and decreasing values south- and north-ward. The O$_3$ meridional transects are characterized by mixing ratio minima of ~ 42-54 ppb at the ITCZ (10-16°S in DJFM and 5-8°N in JJASO) framed by local maxima (~ 53-71 ppb) coincident with the wind shear zones North and South of the ITCZ. O3 gradients are

strongest in the hemisphere of the strongest Hadley cell. IASI UT O$_3$ distributions in DJFM have revealed that the maxima are a part of a crescent-shaped O$_3$ plume above the Atlantic Ocean around the Gulf of Guinea.

CO emitted at the surface is transported towards the ITCZ by the trade winds and then convectively uplifted. Once in the upper troposphere, CO enriched air masses are transported away from the ITCZ by the upper branches of the Hadley cells


and accumulate within the zonal wind shear zones where the maximum CO mixing ratios are found. Anthropogenic and fires both contribute, by the same order of magnitude, to the CO budget of the African upper troposphere.

Local fires have the highest contribution, drive the location of the observed UT CO maxima, and are related to the following transport pathway: CO emitted at the surface is transported towards the ITCZ by the trade winds and further convectively uplifted. Then UT CO enriched air masses are transported away from the ITCZ by the upper branches of the Hadley cells and accumulate within the zonal wind shear zones where the maxima are located. Anthropogenic CO contribution is mostly

from Africa during the entire year, with a low seasonal variability, and is related to similar transport circulation than fire air masses. There is also a large contribution from Asia in JJASO related to the fast convective uplift of polluted air masses in the Asian monsoon region which are further westward transported by the tropical easterly jet (TEJ) and the Asian monsoon anticyclone (AMA).

$O_3$ minima correspond to air masses that were recently uplifted from the surface where mixing ratios are low at the ITCZ.

The $O_3$ maxima correspond to old high altitude air masses uplifted from either local or long distance area of high $O_3$ precursor emissions (Africa and South America during all the year, South Asia mainly in JJASO), and must be created during transport by photochemistry.

This analysis of meridional transects contribute to a better understanding of distributions of CO and $O_3$ in the intertropical African upper troposphere and the processes which drive these distributions. Therefore, it provides a solid basis for

comparison and improvement of models and satellite products in order to get the good $O_3$ for the good reasons.

## 1. Introduction

Tropospheric ozone ($O_3$) has a significant impact on the oxidative capacity of the troposphere by being a major source of hydroxyl radicals (OH) (e.g. Lelieveld et al., 2016; Logan et al., 1981) and also on climate by being a powerful greenhouse gas (e.g. Myhre et al., 2013). The first source of tropospheric ozone is photolysis of $NO_2$, mainly formed by photochemical

cycles involving $NO_x$, $HO_x$ and volatile organic compounds (VOCs) or carbon monoxide (CO) (e.g. Seinfeld and Pandis, 2016). $O_3$ is a secondary compound with a shorter average life time (19 days, Murray et al., 2014) whose contributions are difficult to identify. Indeed, $O_3$ concentrations in the free to upper troposphere (UT) are influenced by (i) transport from the stratosphere (e.g. Hsu and Prather, 2009; Stevenson et al., 2013), (ii) tropospheric transport (e.g. Barret et al., 2016; Sauvage et al., 2005, 2007d; Zhang et al., 2016), (iii) emissions of precursors (VOCs, CO, $HO_2$), (iv) $NO_x$ emissions by surface

combustions but also by free-tropospheric lightning (e.g. Barret et al., 2010; Sauvage et al., 2007c) or by (v) the elimination of $O_3$ by reaction with $HO_2$, OH or photolysis in particular (e.g. Crutzen et al., 1999). Due to its 2-month lifetime (e.g. Edwards et al., 2004), CO is a good tracer of combustion processes (Granier et al., 2011) which emit $O_3$ precursors (i.e. $NO_x$, VOCs, CO). CO impacts the oxidative capacity of the troposphere by being a major sink of OH radicals in non-polluted atmosphere (e.g. Lelieveld et al., 2016) and therefore the climate through enhancement of the lifetime of $CH_4$ the second

most important greenhouse gas (GHG). Oxidation of CO produces $CO_2$ and ozone which are also GHGs (e.g. Myhre et al.,



2013). If it is partly primary (directly emitted into the atmosphere from anthropogenic combustion or forest fires), CO can also be secondary by being formed during the oxidation of VOCs.

The intertropical troposphere is a region of great interest regarding the photochemistry and energy budget of the atmosphere. It indeed combines (i) strong photochemical activity due to intense solar radiation, (ii) high emissions by biomass fires, vegetation and lightning, and (iii) dynamic processes that allow the redistribution of chemical species at regional and global scales. It is therefore an area favorable to the formation of $O_3$. It is also one of the areas of the world where the increase in $O_3$ mixing ratio is most marked since 1980 (Zhang et al., 2016). Africa has the largest continental surface of the intertropical zone and is also a region where high CO mixing ratios (locally above 150 ppb around 250 hPa, e.g. Barret et al., 2010; Cohen et al., 2018; Fu et al., 2016; Sauvage et al., 2007b) are found in the UT. Circulation patterns above intertropical Africa impact the distribution of chemical compounds in the area (e.g. Sauvage et al., 2005, 2007a, 2007c). One of the main circulation patterns is formed by winds from subtropical latitudes to the Equator. They converge at the Intertropical Convergence Zone (ITCZ) before being redistributed vertically thus forming the Hadley cells. The meridional position of the ITCZ has a strong seasonal variability over Africa (e.g. Suzuki, 2011), causing the alternation of dry and wet seasons. This meteorology particularly impacts the $O_3$ and CO distributions in the African UT (e.g. Sauvage et al., 2007b) but also their propagation in the global atmosphere (e.g. Zhang et al., 2016).

Despite the importance of this area, few observations have been carried out and Africa remains little documented in comparison with the northern mid-latitudes. The specific meteorology of the zone makes it difficult for the models to reproduce the concentrations of $O_3$ and CO, notably at high altitude (e.g. Barret et al., 2010; Sauvage et al., 2007a). Satellite observations have been used to document the composition of the African UT but they have a rather coarse vertical resolution (Barret et al., 2008, 2010).

Since 1994, in the framework of the "In-service Aircraft for a Global Observing System" program (IAGOS, previously named MOZAIC; Marenco et al., 1998; Petzold et al., 2015; http://www.iagos.org), airborne measurements of chemical compounds have been made over Africa. Using upper tropospheric measurements of $O_3$ and Relative Humidity (RH) from 1994 to 2004, Sauvage et al. (2007b) showed an ozone minimum collocated with a RH maximum at the ITCZ. Both vertical transport in the ITCZ followed by photochemical production in the upper branches of the Hadley cells were found to contribute to meridional $O_3$ gradient creation. Few CO measurements were available at this time, and none south of the ITCZ.

Between December 2005 and 2013 (2006-2013 hereafter), there were almost daily flights by Air Namibia between Europe and southern Africa, thus providing a unique and highly representative dataset of in-situ $O_3$ and CO on both sides of the ITCZ. The SOFT-IO model (Sauvage et al., 2017c) has been developed to complement the IAGOS database to estimate anthropogenic and fire contributions to CO for each IAGOS measurement. These measurements, together with the SOFT-IO model outputs, allow us to trace the distributions, variability and origins of $O_3$ and CO in the African UT. Since 2008, the IASI instrument onboard the Metop-A satellite also documents UT CO and $O_3$ global distributions (Barret et al., 2016; Tocquer et al., 2015) and therefore complements the African transects provided by IAGOS.



The objectives of this study are to (i) analyze the meridional distribution of O$_3$ and CO mixing ratios observed by IAGOS
      between 2006 and 2013, (ii) explore sources and geographical origins of observed CO anomalies (calculated with the SOFT-
      IO model) and (iii) to give first track on which transport processes drive the CO and O$_3$ meridional transects observed
      especially using retro plumes (calculated with the Lagrangian particle dispersion model FLEXPART) of air masses
      measured by IAGOS. Observational (IAGOS, IASI) and model-based (SOFT-IO) datasets and methods are introduced in

section 2. In section 3, the method to locate the ITCZ and establish the seasonality with IAGOS observations is described
      and validated. In section 4, O$_3$ and CO seasonal meridional transects observed by IAGOS are analyzed and compared to
      IASI observations. In section 5, origins and sources of observed CO are explored with SOFT-IO and, finally,
      characterization of O$_3$ transects is discussed in section 6.

## 2. Data and Methods

**2.1. IAGOS-MOZAIC observations**

      This study is based on the observations made in the framework of the Measurements of OZone, water vapor, carbon
      monoxide and nitrogen oxides by Airbus In-service aircraft (MOZAIC) program (Marenco et al., 1998), now integrated into
      its successor program: IAGOS (https://www.iagos.org, last access: 28 April 2020) (Petzold et al., 2015). In MOZAIC, O$_3$
      and RH have been measured since 1994, and CO since 2002, by airliners of several companies.

In MOZAIC and IAGOS, a dual-beam UV absorption monitor is used to measure O$_3$ with a time resolution of 4 s and
      accuracy estimated at ±2 ppbv /±2 % (Thouret et al., 1998). An improved infrared filter correlation instrument measures CO
      with a time resolution of 30 s and accuracy estimated at ±5 ppbv / ±5 % (Nédélec et al., 2003). Since the setting up of the
      IAGOS program, a single analyzer integrates O$_3$ and CO measurements (Nédélec et al., 2015). In MOZAIC, capacitive
      relative humidity sensors were used to measure RH (MOZAIC Capacitive Hygrometer MCH) (Helten et al., 1998; Neis et

al., 2015a, 2015b; Smit et al., 2014). For IAGOS program, MCH were slightly modified and RH is measured by the IAGOS
      Capacitive Hygrometer (ICH) (Neis et al., 2015b). Both MCH and ICH instruments measure RH with a time resolution from
      1 s at 300 K to 120 s at 200 K and accuracy estimated at ± 5 % (Helten et al., 1998; Neis et al., 2015b). During the 2011–
      2012 overlapping years of MOZAIC and IAGOS programs, systematic inter-comparisons have been performed and have
      demonstrated a good consistency in the both dataset (Nédélec et al., 2015). For convenience and as the IAGOS program

125   database integrates MOZAIC program data, the MOZAIC and IAGOS programs are hereafter commonly referred to as the
      IAGOS program.

      Between December 2005 and November 2013, one Air Namibia aircraft equipped with IAGOS instruments (except in 2010
      for technical reasons) operated daily flights between Europe (London or Frankfurt) and Windhoek in Namibia. It represents
      approximately 2200 flights in the African upper troposphere (between 9 and 12 km altitude) with observations of O$_3$, CO and

130   RH, but also measurements of instant winds (zonal and meridional components, speeds and directions) from the aircraft



navigation instruments. Given its large spatial and temporal coverage, this dataset is unique and offers outstanding opportunities for studying the O3 and CO budget in this region.

Flight trajectories span a wide range of latitudes (between 22°S and 51°N) for a limited variation of longitude (between 17°E and 0°), which allows investigating the meridional distribution of both trace gases.

## 2.2. Complementary data

### 2.2.1. Model-calculated parameters for the IAGOS data points: potential vorticity and SOFT-IO CO contributions

In order to explore origins and sources of CO mixing ratios observed by IAGOS and to characterize the meridional transects, we use the SOFT-IO v1.0 model (Sauvage et al., 2017, http://dx.doi.org/10.25326/2). SOFT-IO v1.0 development and operation are described in details in the reference paper by Sauvage et al. (2017) and SOFT-IO data are freely available in the IAGOS database (https://doi.org/10.25326/3). Briefly, SOFT-IO is a scientific tool which couples FLEXPART 20-day backward plumes of air masses for all the aircraft tracks with global anthropogenic and fire emission inventories for CO, and has been used in various studies (e.g. Cussac et al., 2020; Petetin et al., 2018) to investigate and quantify anthropogenic and biomass burning origin in CO measurements. For the entire IAGOS flight track, SOFT-IO v1.0 estimates the CO contribution (in ppb) of recent ($\leq 20$ days before) worldwide emissions. Estimated contributions are discriminated on the one hand between anthropogenic and fire origins and on the other hand between the 14 different world regions of emissions as defined in Global Fire Emissions Database (GFED, see Fig. 1).

In SOFT-IO v1.0, the version of FLEXPART (Seibert and Frank, 2004; Stohl et al., 2005) is the v9.1 used in backward mode. The estimations of worldwide ground emission contributions to upper-tropospheric measurements depend strongly on the representation adopted for the atmospheric transport and especially for the vertical transport. In our version of FLEXPART, vertical convection is represented using the convective parameterization scheme by Emanuel and Živković-Rothman (1999) as describe by Forster et al. (2007). The global meteorological data for FLEXPART operation are taken from the European Centre for Medium-Range Weather Forecast (ECMWF) analyses (6-hourly) and forecasts (3-hourly), gridded with a 1°×1° horizontal resolution, and 91 vertical levels (137 for year 2013) (Forster et al., 2007; Sauvage et al., 2017c; Stohl et al., 2005).

In this study, the monthly Monitoring Atmospheric Composition and Climate (MACC)/CityZEN EU projects (MACCity) inventory is used for anthropogenic emissions (Diehl et al., 2012; Granier et al., 2011; Lamarque et al., 2010; Van Der Werf et al., 2006) with a longitude–latitude resolution of 0.5°×0.5° and emissions injected in the first altitude level (0–1 km). For fire emissions, the daily Global Fire Assimilation System (GFAS) in its 1.2 version is used (Kaiser et al., 2012) with a 0.1°×0.1° longitude–latitude resolution. As vegetation fires are usually associated with fast updraft, emissions need to be injected at altitude. GFAS inventory integrates injection height data estimated with the Integrated Monitoring and Modeling System for wildland fires parameterization (IS4FIRES, Sofiev et al., 2012) and the Plume Rise Model (PRM, Freitas et al.,



2007) (Rémy et al., 2017). Default injection heights provided in GFAS are used in this study, as described by Sauvage et al. (2017c).

SOFT-IO v1.0 only estimates contributions for anomalies in CO mixing ratios, i.e. primary CO recently emitted ($\leq$ 20 days
before). Background CO, i.e. primary CO emitted for more than 20 days and secondary CO formed by volatile organic compound oxidation, is not calculated.

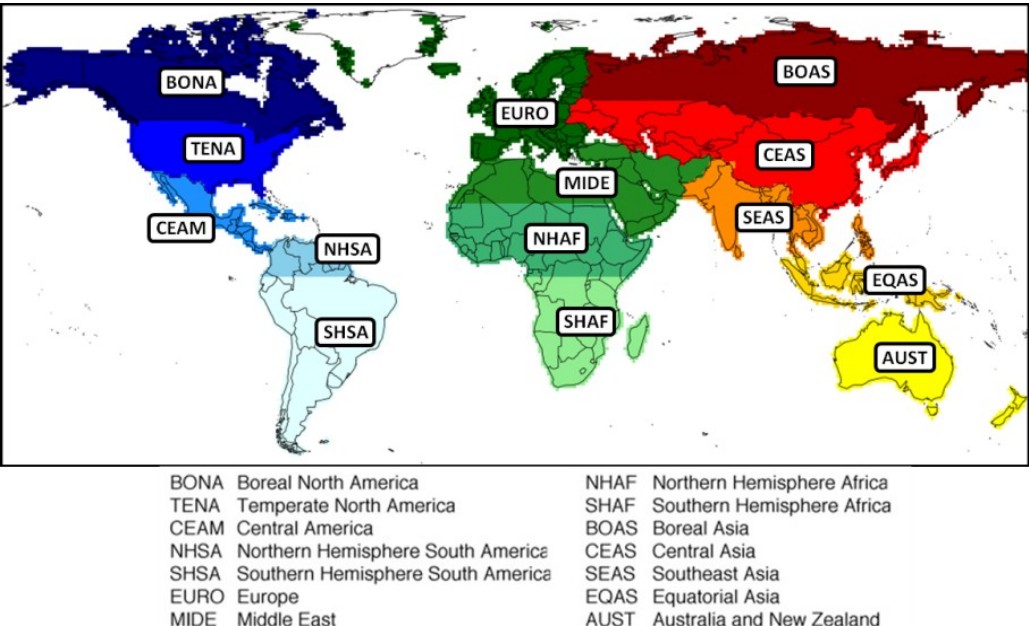

| | | | |
|---|---|---|---|
| BONA | Boreal North America | NHAF | Northern Hemisphere Africa |
| TENA | Temperate North America | SHAF | Southern Hemisphere Africa |
| CEAM | Central America | BOAS | Boreal Asia |
| NHSA | Northern Hemisphere South America | CEAS | Central Asia |
| SHSA | Southern Hemisphere South America | SEAS | Southeast Asia |
| EURO | Europe | EQAS | Equatorial Asia |
| MIDE | Middle East | AUST | Australia and New Zealand |

**Figure 1 : Global Fire Emissions Database (GFED) world regions of emission as used in the SOFT-IO v1.0 model.**

Recently IAGOS database also provides additional ECMWF analysis parameters interpolated along each flight track (http://iagos-data.fr/#CMSConsultPlace:ANCILLARY_DATA, http://www.iagos-data.fr/#L4Place, last access: 28 April 2020). In this study we use potential vorticity to determine stratospheric observations (see Section 2.3). The PV values are calculated with the FLEXPART Lagrangian model using meteorological fields from ECMWF (European Centre for Medium-Range Weather Forecasts) operational analysis, with a 1°×1° resolution, and interpolated along the aircraft flight
path.

### 2.2.2. IASI-SOFRID $O_3$ and CO retrievals

IASI satellite data are used to complement IAGOS observations to document the African UT composition. Indeed, IASI provides daily 2D distributions of UT CO and $O_3$ that put the 1D IAGOS transects into a broader regional context and help documenting the transport processes that control these distributions. The IASI instrument is flying on board the MetOp-A, B
and C satellites launched in 2006, 2012 and 2018 respectively. IASI is a nadir viewing Fourier transform spectrometer



observing the Earth's atmosphere thermal infrared radiation in the 645-2760 cm$^{-1}$ wavenumber region with a spectral resolution of 0.5 cm$^{-1}$ after apodization (e.g. Clerbaux et al., 2009). Metop-A/IASI provides a global coverage twice a day with an overpass time at ~9.30 and ~21.30 local solar time, and a pixel size on the ground of 12 km at nadir. The IASI sensors monitor the tropospheric content of atmospheric trace gases such as $O_3$ (Barret et al., 2011; Eremenko et al., 2008)

and CO (George et al., 2009; De Wachter et al., 2012). In the present study, we use data retrievals performed by the Software for a Fast Retrieval of IASI Data (SOFRID) (Barret et al., 2011; De Wachter et al., 2012).

Vertical profiles are retrieved from IASI cloud free spectra on 43 fixed pressure levels from the surface to 0.1 hPa. Barret et al. (2011) have shown that IASI enabled the independent retrieval of $O_3$ in the lower-middle troposphere (surface-225 hPa) and in the upper troposphere - lower stratosphere (UTLS, 225-70 hPa). The agreement with $O_3$ sonde data is especially good

for the UTLS column with correlation coefficients of 0.8 (resp. 0.95) and biases of 17.5+/-20% (resp. 10+/-10%) in Dufour et al. (2012) (resp. Barret et al. (2011)). According to De Wachter et al. (2012) SOFRID-CO retrievals are also providing 2 independent pieces of information, one in the lower and the other one in the upper troposphere. The SOFRID-CO data have been validated against IAGOS-MOZAIC airborne in-situ data (De Wachter et al., 2012). SOFRID-CO retrievals are able to capture the seasonal variability of CO at mid-latitudes (Frankfurt) as well as at tropical latitudes (Windhoek) in the lower

(upper) troposphere with correlation coefficients of 0.85 (0.70). At Windhoek, in the lower (upper) troposphere, SOFRID-CO data are biased low with 13+/-20% (4+/-12%). SOFRID-CO and $O_3$ data have been used to document the composition of the UTLS Asian monsoon anticyclone (AMA) in Barret et al. (2016). In the present study, we use monthly averaged SOFRID-CO and $O_3$ retrievals on a 1° latitude × 1° longitude grid. IASI data used span on the 2008-2013 period.

### 2.2.3. NCEP/NCAR reanalysis data

In order to discuss and validate the methodology applied to locate the ITCZ from IAGOS meteorological data, NCEP reanalysis data of RH, winds and outgoing longwave radiation (OLR) were used (Kalnay et al., 1996; https://www.esrl.noaa.gov/psd/cgi-bin/data/composites/printpage.pl, last access: 28 April 2020). NCEP reanalysis data are global analysis of atmospheric fields combining measurements (land surface, ship, rawinsonde, aircraft, satellite, etc.) with modeling data, using a data assimilation system covering a large period from 1957 to today.

### 2.3. IAGOS and SOFT-IO data treatment

This study focuses on the African UT. To eliminate lower tropospheric data, only high-altitude observations recorded above 300 hPa are selected (thus removing observations collected during takeoffs and landings). In the intertropical latitudes, only tropospheric air masses are encountered due to the high altitude of the tropopause but in the extratropical latitudes, both tropospheric and stratospheric air masses can be encountered. To discriminate upper tropospheric and lower stratospheric

data we use the dynamical tropopause defined as the isosurface of 2 PVU (Potential Vorticity Unit) as in Thouret et al. (2006) and recently used in Cohen et al. (2018). The tropopause is actually considered as a 30 hPa thick layer, centered on the 2 PVU isosurface. To consider only upper tropospheric data, observations obtained at a pressure greater than the 2 PVU



pressure + 15 hPa are eliminated. Finally, data are aggregated and averaged by degree of latitude on the entire meridional transect. Data are also averaged for different time periods as described in details in the following subsection. Table 1 presents the number of IAGOS flights with $O_3$, CO or RH data available for each season of the studied period (see section 3 for precisions on the season's delimitation methodology).

**Table 1 : list of available IAGOS data per season (see section 2.4 for more details on season definition)**

|  |  | Number of IAGOS flights | with $O_3$ data | with CO data | with RH data |
|---|---|---|---|---|---|
| **2006** | **DJFM** | 121 | 120 | 119 | 121 |
|  | **AM** | 60 | 59 | 59 | 60 |
|  | **JJASO** | 149 | 139 | 147 | 149 |
|  | **N** | 30 | 17 | 30 | 30 |
| **2007** | **DJFM** | 121 | 112 | 114 | 121 |
|  | **AM** | 50 | 49 | 50 | 50 |
|  | **JJASO** | 128 | 128 | 128 | 128 |
|  | **N** | 30 | 30 | 30 | 30 |
| **2008** | **DJFM** | 74 | 74 | 74 | 74 |
|  | **AM** | 65 | 65 | 46 | 65 |
|  | **JJASO** | 151 | 116 | 151 | 151 |
|  | **N** | 31 | 0 | 31 | 31 |
| **2009** | **DJFM** | 113 | 98 | 113 | 113 |
|  | **AM** | 61 | 25 | 59 | 61 |
|  | **JJASO** | 127 | 0 | 47 | 127 |
|  | **N** | 15 | 0 | 0 | 15 |
| **2011** | **DJFM** | 86 | 36 | 36 | 86 |
|  | **AM** | 53 | 23 | 51 | 53 |
|  | **JJASO** | 138 | 102 | 133 | 138 |
|  | **N** | 25 | 24 | 24 | 25 |
| **2012** | **DJFM** | 101 | 83 | 36 | 101 |
|  | **AM** | 50 | 50 | 42 | 50 |
|  | **JJASO** | 140 | 109 | 134 | 140 |
|  | **N** | 34 | 19 | 34 | 30 |
| **2013** | **DJFM** | 33 | 32 | 32 | 12 |
|  | **AM** | 41 | 0 | 0 | 40 |
|  | **JJASO** | 149 | 0 | 110 | 149 |
|  | **N** | 0 | 0 | 0 | 0 |

## 3. Seasonality of the African upper tropospheric circulation

Ozone precursors are emitted in large quantities across the African continent (Liousse et al., 2014; Sauvage et al., 2007d), but our understanding of the transport and effects of these precursors is limited by a lack of in situ data in the African UT. . The climate of southern and northern Africa is characterized by alternating wet and dry periods which are modified by the position of the ITCZ. Within the ITCZ warm and humid surface air masses converge and are convectively uplifted into the





upper troposphere. The position of the convergence zone gives rise to the "wet" seasons. The uplifted air masses are then advected polewards in the upper branches of the Hadley cells. In the upper troposphere, the air masses are zonally transported by (i) the northern and southern subtropical jets, (ii) the flow from the South Atlantic anticyclone and (iii) the easterly winds of the tropical easterly jet (TEJ) and Asian monsoon Anticyclone (AMA) (in boreal summer) (e.g. Krishnamurti et al., 2013). The dry air in the descending branches of these cells creates the conditions for wildfires and the

resulting emission of ozone precursors. Seasonality in Africa is thus closely related to the geographical shift of the ITCZ from the northern hemisphere during the boreal summer to the southern hemisphere during the boreal winter. The classic four seasons of three months is therefore not adapted to our vast area of study centered on the African inter-tropical zone and it was therefore necessary to establish a new definition of seasons.

### 3.1. Seasonality of the position of the ITCZ according to IAGOS data

In the UT the ITCZ is characterized by (i) high levels of RH, (ii) the predominance of easterly zonal winds and (iii) the divergence of meridional winds. We use the meteorological fields (RH and zonal winds) measured by the IAGOS aircraft along with NCEP reanalysis (Section 3.2) to locate the position of the ITCZ and to identify seasonal regimes which we will use in later sections. The meridional monthly transects of winds and RH were analyzed to locate the latitude of the ITCZ during each month of the studied years. Seasons were then defined by grouping monthly data presenting an internal

consistency for each parameter: zonal and meridional winds and RH. This analysis allowed us to define two main seasons:

- From December to March (DJFM), during which the position of the ITCZ along the trajectory of IAGOS flights is relatively stable between 14.5°S and 4.5°S,
- From June to October (JJASO), during which the position of the ITCZ is relatively stable between 4.5°N and 9.5°N,

And two so-called transition periods:

- From April to May (AM), during which the ITCZ position shifts to the north,
- The month of November (N), during which the ITCZ position shifts to the south.

We use these definitions in the following sections. Figure 2 shows the IAGOS inter-tropical seasonal mean transects of zonal winds (Fig. 2.a, b, c and d), meridional winds (Fig. 2.d, e, f and g) and RH (Fig. 2.m, n, o and p). Inter-tropical transect

boundaries are fixed as (i) 20°S in the southern hemisphere (latitude of the first measurements at high altitude in the south); (ii) the average latitude at which the flights cross the tropopause in the northern hemisphere (blue lines in Fig. 2.i, j, k and l). This latest limit ranges between 34 and 38°N in JJASO and between 26 and 28°N in DJFM.

The zonal wind transects display a low interannual variability which allows us to clearly locate the zone of easterly wind (zonal winds < 0) framing the ITCZ at each season (colored rectangles in fig. 2.i, j, k and l). Zonal winds < 0 are found from

18°N and 8°S in JJASO and between 7°N and 18°S in DJFM. With lower intensity, meridional winds are used to locate the divergence latitude of meridional winds in the easterly wind zone of the ITCZ: at ~8°N in JJASO and ~7°S in DJFM (black lines in fig. 2.i, j, k and l). The Hadley cells are known to be non-symmetric about the ITCZ with the winter cell stronger



and broader than the summer cell (Tanaka et al., 2004). Here, observations are in agreement with this statement with stronger meridional winds north of the ITCZ in DJFM (~ 2 m s$^{-1}$ versus 6 m s$^{-1}$ respectively 10 ° south and north of the

meridional wind divergence latitude) and south of the ITCZ in JJASO (~ 4.5 m s$^{-1}$ versus 2.5 m s$^{-1}$ respectively 10 ° south and north). For the transition periods, the location of the mean divergence does not really make sense because of the shift of the ITCZ during these periods. However, we can observe the thinning of the easterly wind band and the decrease of wind intensity inside of the band during April-May (easterly wind band ranging from 8°N at 7°S with mean zonal winds intensities < 5 m s$^{-1}$, Fig. 2.b).

The transects of average RH with respect to liquid water show low interannual variability for the three DJFM, April-May and JJASO periods. For the period of November a large variability is observed and can be explained in part by the limited number of data and the transitory nature of the period (i.e. with fast variation in meteorology). The transects of the main seasons show maxima of RH located at the ITCZ as well as a new increase at the northern latitudes of the studied area. The RH maxima and therefore the ITCZ are located between 14.5°S and 4.5°S in DJFM and between 4.5°N and 9.5°N in JJASO.

For the April-May transition period, two mean RH peaks are observable due to the northward displacement of the ITCZ during this period. The location of the ITCZ for the two main seasons is shown in Fig. 2. In the next section, the IAGOS data along linear flight tracks are compared with NCEP reanalysis to offer a broader meteorological context.

Figure 2 : IAGOS meridional and seasonal transects of zonal winds (a, b, c and d), meridional winds (e, f, g and h) and relative humidity (m, n, o and p). Panels i, j, k and l are schematic representations of average easterly winds zone latitudes (colored rectangles), average latitude of meridional wind divergence (black lines) and average latitude of the intersection between flight trajectories and the tropopause (northern limit of tropospheric measurements). The vertical blue shaded areas represent the deduced ITCZ position.





### 3.2. Consistency with NCEP/NCAR reanalysis data

Figures 3.e,f,g and h show the mean RH at 300 hPa above Africa for the four seasons, as defined above, between 2006 and 2013 from NCEP reanalysis data along with the all the individual IAGOS flight tracks during the same period. The NCEP reanalysis shows a zone of high RH (RH > 60%) framed by two areas where the RH < 22.5% whose positions change during the year. In JJASO (Fig. 3.g), the humid area is located on the northern edge of the Gulf of Guinea (~10° N to 2°S on the flight trajectories). In DJFM (Fig. 3.e), the humid area is located on the southeastern edge of the Gulf of Guinea (~0° to 15°S). These locations of the high RH, associated with the ITCZ, are in agreement with IAGOS observations (Fig. 2.m and o). The seasonal evolution of the RH amplitude on the flight trajectories is also in agreement with IAGOS with higher amplitude in JJASO than in DJFM due to the shifting of the low RH areas east and west from the flight tracks (Fig. 3.e).

Between these two seasons, in April-May (Fig. 3.f) and November (Fig. 3.h), the position of the different humid and dry zones in the upper troposphere correspond to intermediate situations. In April-May, it is interesting to note that the RH range appears to be lower than the rest of the year (20% < RH < 60%) with less marked and extensive dry areas and a small humid area. This is partly explained by the rapidly changing meteorological conditions over this period (e.g. Suzuki, 2011) and its two-months average in Fig. 3.f.

Figures 3.a,b,c and d show the mean circulation at 250 hPa above Africa for the four seasons (2006-2013) from NCEP reanalysis. In JJASO (Fig. 3.c), the zone of easterly winds is at its northernmost position with zonal wind inversion latitudes around 17°N and 7°S on the trajectory of IAGOS flights and the inversion of the meridional winds are observed around 10°N, locating the ITCZ at the same latitude than with IAGOS data. In DJFM (Fig. 3.a), the easterly wind zone is at its southernmost position with zonal wind inversion latitudes around 5°N and 17°S and the inversion of the meridional winds are observed around 5°S, in agreement with in situ measurements.

Between these two seasons, in April-May (Fig. 3.b), we observe that the position of the easterly wind zone lies as expected, between that of DJFM and that of JJASO. This band of easterly winds is thinner (15° compared with 24° in JJASO and 22° in DJFM) and has a lower average wind speed (< 5 m s$^{-1}$) compared with main seasons as observed with IAGOS (Fig. 2.b).

Finally, the OLR is also a good indicator of the position of the ITCZ. Deep convective clouds present at the ITCZ are associated with OLR below 220 W m$^{-2}$ (e.g. Park et al., 2007). These areas are represented in Fig. 3 by black hatched contours. We observe that these areas above Africa correspond to high RH zones and latitudes dominated by easterly winds where the meridional component is close to 0. All these NCEP reanalysis data corroborate IAGOS observations and validate previous methodology developed to locate the ITCZ with IAGOS in situ measurements.



**Figure 3 : Seasonal mean wind circulation at 250 hPa (a, b, c and d) and mean relative humidity at 300 hPa (e, f, g and h) above Africa between 2006 and 2013 from NCEP reanalysis. Red lines represent the IAGOS flight trajectories. Black hatched contours represent areas with OLR < 220 W m$^{-2}$.**

## 4. O$_3$ and CO meridional transects in the African upper troposphere

### 4.1. IAGOS observations

Figure 4 shows IAGOS average meridional transects of O$_3$ (Fig. 4.a,b,e and f) and CO (Fig. 4.c,d,g and h) for the four seasons defined in section 3.1. The study focuses on DJFM and JJASO seasons. Indeed, the analysis and interpretation of the transects for the two transition periods (noticed AM and N) is difficult because of the important meteorological variations that occur during these transition periods. Table 2 summarizes the locations and extremum levels of the meridional O$_3$ and CO transects seen by IAGOS.

For DJFM and JJASO, O$_3$ meridional transects are characterized by a minimum above the ITCZ and positive gradients directly north and south of the ITCZ in agreement with Sauvage et al. (2007b) who used a different definition for the seasons and less frequent measurements. The transects show (i) no clear interannual variability in terms of gradients and location of



the extrema but (ii) an important mixing ratio variability with about 15 ppb difference between years with the highest $O_3$ level (2013 for DJFM and 2008 for JJASO) and years with the lowest ones (2007 for DJFM and JJASO). In this section, we calculate a gradient in ozone between the observed minimum and the northern or southern maximum. The calculation of the

gradients gives an indication of ozone build-up along meridional (Hadley cells) and zonal transport.

In DJFM, we observe a sinusoidal pattern. For each year, the minimum is found between 10-16°S co-located with the ITCZ. There is an interannual variability of about 15ppb between the the year with the lowest mixing ratio (2007) and the year with the highest mixing ratio (2013) at the ITCZ.  The maximum ozone mixing ratios are found (i) north of the ITCZ between 5-10°N leading to a gradient north of the equator of $\approx$ 0.85 ppb deg$^{-1}$ starting from the ITCZ minima and (ii) south of the ITCZ

around 20°S leading to a gradient south of the equator of $\approx$ 1.2 ppb deg$^{-1}$. The maximum around 20°S at the limit of our domain is confirmed by the IASI data which shows a decrease of $O_3$ south of this latitude (see Fig. 4.a). The gradients north of the equator are much higher than that observed in  Sauvage et al. (2007b) ($0.34 \pm 0.07$ ppb deg$^{-1}$ north of the ITCZ during boreal winter. There was no data south of the equator reported in Sauvage et al 2007b).

In JJASO (fig 4b), ozone minima are observed around 5-8°N (ITCZ position). Mixing ratios increase with increasing

distance from ITCZ, leading to 0.95 (1.05) ppb deg$^{-1}$ gradients in the southern (northern) hemisphere, compared with lower values in JJA in Sauvage et al. (2007b) study 0.62 (0.94) ppb deg$^{-1}$ in south (north) hemisphere.



**Figure 4 : IAGOS meridional and seasonal transects of ozone (a, b, e and f) and carbon monoxide (c, d, g and h). The vertical blue**
**rectangles represent the deduced ITCZ position. The grey areas represent the IASI data range at 253 hPa on the flight trajectories**
**for the 2008 to 2013 period (no data available before 2008).**





CO meridional transects are characterized by a maximum above the dry region 10° north (south) away from the ITCZ in
DJFM (JJASO), and negative gradients directly north and south of the maxima. Similarly to $O_3$, CO transects show no clear
interannual variability of the steepness of the gradient and locations of the extrema but an important variability of the
average mixing ratios with up to 30 ppb difference between years with the highest CO level (2007 for DJFM and 2008 for
JJASO) and years with the lowest one (2013 for DJFM and 2011 for JJASO). In this section, CO gradients are calculated
between the observed maximum and the northern or southern limit. They are thus both negative and are respectively noticed
south CO gradient and north CO gradient.

In DJFM, maximum CO mixing ratios are reached around 0-5°N. CO mixing ratios decrease south (north) of the ITCZ with
a gradient of ≈ -2.0 (-1.7) ppb deg$^{-1}$ (2011, 2012 and 2013 are not considered because of the few amount of CO data).

In JJASO, the highest CO mixing ratios are located around 3-7°S. The south (north) CO gradient is ≈ -3.3 (-1.7) ppb deg$^{-1}$
(2009 is not considered in this gradient calculation because of the few amount of CO data for this year at JJASO). During the
two transition periods in April-May and November, the presence of two peaks on the CO transects highlights the transition
phase of the position of the ITCZ.

During the two seasons, CO maxima are located in the hemisphere of the strongest Hadley cell, which seems to have a
strong impact on CO distributions in the African upper troposphere.

**Table 2 : Synthesis of the characteristics of meridional distributions of CO and $O_3$ in the African upper troposphere as seen by**
**IAGOS.**

| Season | $O_3$ Max. | | $O_3$ Min. | | $O_3$ Gradients | | CO Max. | | CO Gradients | |
|---|---|---|---|---|---|---|---|---|---|---|
| | location | mixing ratio | location | mixing ratio | north | south | location | mixing ratio | north | south |
| DJFM | ≈ 20°S | 53 - 65 ppb | 10-16°S | 42 - 54 ppb | 0.85 ppb deg$^{-1}$ | 1.2 ppb deg$^{-1}$ | 0-5°N | 132 - 165 ppb | -1.7 ppb deg$^{-1}$ | -2.0 ppb deg$^{-1}$ |
| | 5-10°N | 56 - 71 ppb | | | | | | | | |
| JJASO | 15-20°S | 64 - 80 ppb | 5-8°N | 43 - 54 ppb | 1.05 ppb deg$^{-1}$ | 0.95 ppb deg$^{-1}$ | 3-7°S | 128 - 149 ppb | -1.7 ppb deg$^{-1}$ | -3.3 ppb deg$^{-1}$ |
| | 32-38°N | 73 - 87 ppb | | | | | | | | |

## 4.2. IASI – IAGOS transect comparison

In a similar way to the previous section where we used the NCEP reanalysis of wind and humidity to put the IAGOS flight
tracks in a meteorological context, here we use IASI satellite observations of $O_3$ and CO to help put the IAGOS observations
in a chemical context. We compare $O_3$ and CO transects obtained with IAGOS with the two-dimensional fields of $O_3$ and CO
from IASI which give a wider view over Africa and the adjacent oceans allowing us to see the influence of regional
circulation patterns on the distribution of CO and $O_3$.The IASI data were processed to take into account only a linear corridor
of 5° width (longitude) at a pressure level of 253 hPa, including all the trajectories of IAGOS flights (the central trajectory of
this corridor follow the equation: lon = 0.14° × lat + 12.14° to include all flights). IASI data used span on the 2008-2013



period. The IASI data also represent a vertical thickness of ~30 hPa due to the vertical sensitivity of the satellite. This leads to some important difference compared with the IAGOS data as described below.

The grey shaded areas in Fig. 4 represent the IASI data range at 253 hPa on the flight trajectories between 2008 and 2013, i.e. the range between the lowest and the highest seasonal mean mixing ratio observed by IASI during the 2008/2013-period

for each latitude degree. North of the intertropical zone, $O_3$ mixing ratios rise sharply because of the lower tropopause and the sampling of stratospheric air by the satellite sensor. The IASI $O_3$ data for the two main seasons are in good agreement with IAGOS. They correctly capture (i) the wave profile in DJFM, (ii) the minimum and the two positive gradients in JJASO and (iii) the orders of magnitude of the mixing ratios. However, the meridional location of some extrema is not in phase with IAGOS. In DJFM, the minimum of $O_3$ at the ITCZ and the southern maximum are consistent with IAGOS. The northern $O_3$

maximum is shifted 8-10° southwards relative to IAGOS with $O_3$ mixing ratios similar to IAGOS measurements (≈ 55-67 ppb). South of the study domain, IASI retrievals complete the IAGOS observations documenting a -1.35 ppb deg$^{-1}$ decrease in $O_3$ mixing ratio. In JJASO, IASI $O_3$ minimum is shifted by 10° northwards relative to IAGOS with consistent mixing ratios and gradients. South of the local maximum at ~ 8°S, IASI $O_3$ decreases.

IASI CO is highly consistent with IAGOS with maxima located at similar latitudes for the four seasons. However, the

maximum mixing ratios observed with IASI are systematically biased low relative to those measured by IAGOS by 20 ppb on average (up to 40 ppb in DJFM). This results in lower CO gradients with IASI. For the four seasons, IASI observations beyond IAGOS domain indicate that (i) CO decreases uniformly between 20°S and 30°S and (ii)  CO remains almost constant between 30 and 50°N.

IASI $O_3$ and CO retrievals have been shown to be sensitive to the UT but with a coarse vertical resolution of about 6 kms

(see Barret et al., 2011 for $O_3$ and De Wachter et al., 2012 for CO). Some of the discrepancies and observed biases are related to the very different nature of both types of observations. However, the $O_3$ and CO African UT latitudinal transects documented by IASI and IAGOS are overall reasonably consistent highlighting that IASI can successfully complement IAGOS to obtain horizontal distributions.

Figure 5 shows the mean 2008-2013 seasonal CO (a and b) and $O_3$ (c and d) distributions observed by IASI at 253 hPa for

the two main seasons. These maps complement the latitudinal information given by IAGOS about the regional CO and $O_3$ distributions.

For CO, we observe the highest mixing ratios above west and central Africa in DJFM (Fig. 5.a) and over central and southern Africa in JJASO (Fig. 5.b). In both seasons, there is a strong outflow of CO over the gulf of Guinea. The comparison of the IASI CO distributions (Fig 5.a and b) with the wind distributions (Fig 3.a and c) highlights the

accumulation of CO within the zonal wind shear zones north and south of the ITCZ in DJFM and JJASO respectively.

As noted above (Fig. 4) the IASI CO maxima are consistent with those observed by IAGOS along the latitudinal transects. In addition, in DJFM, more localized and less important hotspots of CO concentrations are also detected throughout the northeast quarter of Africa and the south of the Arabian Peninsula.



For $O_3$, in JJASO, we observe a maximum over south western Africa, spreading westwards over the Gulf of Guinea and the
Atlantic and eastwards over the south-western Indian Ocean. In DJFM, the two $O_3$ maxima observed by IAGOS correspond
to the two branches of a crescent-shaped high $O_3$ plume located above the Atlantic. The $O_3$ minimum at the center of the
crescent coincides with the RH minimum at the ITCZ (Fig 3). The two branches of the $O_3$ maxima are located within the
wind-shear zones with winds minima on both sides of the ITCZ (see winds on Fig. 3).


**Figure 5 : Seasonal mean IASI CO (a and b) and $O_3$ (c and d) mixing ratios at 253 hPa between 2008 and 2013. The points
represent IAGOS measurements averaged on all studied years.**

## 5. Origin and sources of CO

Figure 6 represents, averaged between 2006 and 2013, IAGOS CO for the four seasons (Fig. 6.a to d) and the contributions
to these CO mixing ratios from anthropogenic and wildfire sources (Fig 6.e to h) as estimated with SOFT-IO. These
anthropogenic and wildfire sources are further decomposed by region in figures 6.i to l and 6.m to p respectively. The





different regions are denoted by their GFED acronym (see fig 1). It is worth recalling that the background is not simulated by SOFT-IO (see section 2.2.1).

The meridional transects of total CO anomalies from SOFT-IO exhibit similar behavior to the IAGOS data with CO mixing
ratios characterized by a clear maximum shifting from the north of the ITCZ in DJFM to the south of the ITCZ in JJASO. In DJFM, the fire and anthropogenic contributions shown by SOFT-IO are in phase, with the maximum CO, as with the IAGOS observations, found at the equator. In JJASO, the maximum contribution from fires and the maximum observed CO both occur at 5°S while the maximum contribution from anthropogenic sources is located at 12°N. The resulting total contribution has therefore shifted north (~Equator) relative to the observed one. This may indicate an under estimation of SHAF fire
and/or anthropogenic contributions, or an overestimation of the anthropogenic contribution from the SEAS and CEAS regions which are responsible of the high CO North of the Equator (see fig. 6.k).

The absolute quantitative CO mixing ratios estimated with SOFT-IO are subject to biases and uncertainties especially due to uncertainties in emission inventories and difficulties in representing tropical deep convection either by the ECMWF analyses or by the FLEXPART model. However, the comparisons between the different estimated contributions remain consistent.
The anthropogenic and fire CO sources are of the same order of magnitude during the main seasons while the anthropogenic source dominates during the transition seasons due to a decrease in fire contributions. Overall therefore, the contribution of fires is not the dominant contribution to CO in the African upper troposphere consistent with the observed and continuous increase in anthropogenic emissions (e.g. Liousse et al., 2014), and the slight decrease of fire emissions (GFAS, Kaiser et al., 2012).
The anthropogenic contributions are mainly local with NHAF and SHAF representing about 66 to 90% of the estimated anthropogenic contributions at the location of the CO maxima. The local contributions have rather low seasonal variabilities. As expected SHAF contributes mostly south of the Equator (maximum between 5° and 10°S) and NHAF north of the Equator (maximum around 10°N). The CO emitted locally at the surface is transported in the lower troposphere by the trade winds towards the ITCZ, and then uplifted to UT (Edwards et al., 2003; Sauvage et al., 2006). Thus, in DJFM (JJASO), the
maximum contribution of SHAF (NHAF) is co-located with the position of the ITCZ. In addition to local influence, anthropogenic CO in the African UT is influenced by long-range transport from Asia. The contribution from southeast and central Asia (SEAS and CEAS) has an important seasonal variability. The SEAS anthropogenic CO contribution is negligible in the period April-May but represents up to 30% of total CO anthropogenic contribution in DJFM and 21% in November at the maximum located at the Equator. The Asian contribution is the strongest in JJASO in the northern
hemisphere representing up to 66% of estimated anthropogenic contributions at 20°N. Barret et al. (2008) have shown that this strong Asian contribution results from south Asian emissions uplifted to the UT by the deep convection associated with the monsoon, and then transported westward by the Tropical Easterly Jet. We also observe an important contribution from the Middle East (MIDE) in JJASO, centered at the ITCZ. This MIDE contribution corresponds to surface air masses from North Africa and Arabia which followed the same circulation as local surface air masses to UT ITCZ. In the northern



hemisphere, a small American (TENA and CEAM) contribution transported via upper-level westerly winds close to the subtropical jet is observable during all the seasons.

The fire contributions are predominantly local (NHAF and SHAF) for all the seasons with an important seasonal variability. The geographical origin of CO from fires follows the dry and wet seasons of the two hemispheres. During the dry season of the northern (southern) hemisphere, NHAF (SHAF) fires contribute more than 90% to the meridional CO transects. The

maximum contributions (≈13-14 ppb, i.e. in addition to the background) are shifted by approximately 10° towards the dry zone relative to the ITCZ. This is explained by the location of the emission zones (i.e. dry zones) which are shifted relative to the ITCZ. CO emitted at the surface is transported at low altitudes towards the ITCZ by the northeast trade winds and the Harmattan in DJFM and by the southeast trade winds and the south Atlantic anticyclone in JJASO (Edwards et al., 2003; Sauvage et al., 2006). The air masses are then uplifted by deep convection and move away from the ITCZ in the upper

branches of the Hadley cells (e.g. Barret et al., 2010; Sauvage et al., 2007c). The contribution to the CO from fires in the transition periods (maxima ≈ 5 ppb in April-May and ≈ 9 ppb in November above background levels) is mostly from SHAF in the southern hemisphere and from NHAF in the northern hemisphere. The contributions from fires are lower during these transition periods due to the moisture retained from the previous wet season, or the increasing humidification going into the next wet season. Small contributions from Indonesia (EQAS) are observed in JJASO (maximum relative contribution ≈10%

at 20°N) and in November (≈20% at 0°N). They correspond to air masses advected by the TEJ and diffused north and south during the transport. South American (SHSA) contributions are also observed in JJASO in the southern hemisphere (maximum relative contribution ≈30% at 20°S) and in November on the entire transect (maximum relative contribution ≈50% at 20°S). The polluted air masses are transported in the UT by the westerly winds of the south Atlantic anticyclone in the southern hemisphere and by the subtropical jet in the northern hemisphere.

It is interesting to note that, in both DJFM and JJASO, the CO plumes are transported over the Gulf of Guinea by the easterly winds that dominate within the ITCZ (see Fig. 3) as documented by IASI (Fig. 5). The IASI distributions also highlight that during JJASO the strong TEJ (Fig 3.c) transport the CO plume westwards while during DJFM the transport is north-westward due to a northwards wind component over Africa (Fig 3 a).


**Figure 6 : Meridional and seasonal transects of CO measured by IAGOS (a, b, c and d), CO total anthropogenic and fire contributions calculated by SOFT-IO (e, f, g and h), anthropogenic CO regional origins estimated with SOFT-IO (i, j, k and l) and fire CO regional origins estimated with SOFT-IO (m, n, o and p). The vertical blue rectangles represent the deduced ITCZ position. IAGOS and SOFT-IO data shown are averaged on all studied years (end 2005-2013).**




## 6. Origins of $O_3$ and associated transport pathways

In the previous section we used the SOFT –IO tool to attribute high concentrations of CO to anthropogenic and wildfire sources across different regions.  As tropospheric $O_3$ is a secondary pollutant, the source attribution is more complex, as the

ozone may be lost or produced along its trajectory from the surface. There may be local sources in the upper troposphere such as intrusions of ozone rich air from the stratosphere, and photochemical production from $NO_x$ generated by lightning, or even via $NO_x$ emitted by aircraft (which may be relevant????? should maybe mention this). Analysis of the meridional transects of ozone is more complex. It is possible to determine the origin of air masses and the spatial variability of ozone and give the associated transport pathways, independently of CO mixing ratio.  Therefore in this section we do not use the

SOFT-IO as we did for CO but we look at the FLEXPART backward plumes initiated at the IAGOS data points. . Based on previous modeling studies (Sauvage et al., 2007a, 2007c, 2007d), we hypothesize that $O_3$ is produced during the transport of air masses impacted by precursors emitted  at the surface (such as CO as described in the previous section) and sources at altitude such as lightning as inferred in Sauvage et al. (2007d) study with a chemistry and transport model.

Figures 7 and 8 represent the FLEXPART backward plumes corresponding to air masses characterized by $O_3$ maxima and

minima as observed by IAGOS in DJFM (Fig. 7) and JJASO (Fig. 8). The backward plumes are displayed as the integration of 20-day residence times (equivalent to the origin probability density, see Stohl et al., 2005) summed vertically and averaged over the whole period (end 2005-2013). For each case, backward plumes are computed for sections of 5° of latitude from the corresponding IAGOS flights.

In DJFM, the $O_3$ minimum is observed at the ITCZ ($\approx$ 11°S, see Fig. 4) and corresponds to African air masses (Fig. 7.e)

rising from the surface (Fig. 7.d and 7.f). $O_3$ mixing ratios are globally lower in surface air masses due to either clean maritime air masses, or $O_3$ titration in fresh polluted air masses. The area of origin of the air masses (Fig. 7.e) corresponds to the convergence zone delimited by the high RH and the OLR below 220 W m$^{-2}$ in Fig. 3.e. We note that the ascension of the surface air masses takes place to the south-east of IAGOS aircraft, i.e. in the center of this convergence zone, where the Hadley cell vertical wind speed is the strongest.

The maximum of $O_3$ observed by IAGOS in DJFM ($\approx$ 10°N, see Fig. 4) corresponds to high altitude air masses transported during the previous 20 days (Fig. 7.a to 7.c). These are:

     (i)     air masses uplifted at the ITCZ and transported away from the ITCZ by the Hadley cell upper branches . Chemical processing during transport allowed $O_3$ production from African LiNOx emitted in the UT at the ITCZ and from uplifted African surface emissions (CO maxima at $\approx$ 2°N).

(ii)    air masses uplifted in South America and transported by the high altitude westerly winds located north of the ITCZ with similar chemical processing from precursors emitted by fires and by lightning.

As already discussed (section 4.2), in DJFM there are two $O_3$ maxima documented by IASI on each side of the ITCZ which stem from the same crescent-shaped plume (Fig. 5.c). The northern African $O_3$ maximum is located a bit to the South relative to the CO maximum according to IASI (Fig 5.a,c and 4.a,c) and a bit further North according to IAGOS (Fig. 4.a and





c). Nevertheless, in both cases they are located on the same side of the ITCZ. The FLEXPART run indicates that at the location of the northern $O_3$ maximum seen by IAGOS, air masses are mostly coming from the West African fire region explaining the coincidence between the $O_3$ and CO plumes. The air masses observed by IAGOS at 20°S (local maximum of $O_3$) may have been thus transported in the UT from the Gulf of Guinea to the South Atlantic Ocean and then to the south of Africa where they have been sampled by IAGOS. Mixing with surrounding $O_3$-poorer air masses during this transportation is

probably compensated by the input and photochemistry of precursors from South America and lightning.

In JJASO, the minimum of ozone observed by IAGOS at the latitude of the ITCZ ($\approx$ 6°N, see Fig. 4) corresponds mainly to $O_3$-poor African air masses (Fig. 8.e) convectively uplifted from the surface (Fig. 8.d and 8.f) and transported westward during the convection. Clean air masses from the Indian Ocean region convectively uplifted and transported by the TEJ also contribute to the $O_3$ minimum. During this season the $O_3$ maximum ($\approx$ 15°S, see Fig. 4) corresponds to air masses which

have been uplifted at the ITCZ in Africa (northeast of the aircraft position) or in South America more than twenty days before the observations. (Fig. 8.a and 8.c). The coincidence between the CO and $O_3$ plumes from IASI (Fig. 5.b and d) indicates that the $O_3$ production is directly impacted by African emissions.

North of 20°N, a decrease in $O_3$ mixing ratios in DJFM is observed, which is not observed in JJASO (see Fig 4). To understand this difference, the FLEXPART backward plumes corresponding to IAGOS flights between 20° and 25°N for the

two seasons are presented in Fig. 9. In the two cases, the air masses mainly resided above 9000 m in the 20 days preceding the measurements. In DJFM (Fig. 9.a to 9.c), the low $O_3$ mixing ratios correspond to high altitude old and clean air masses transported by the subtropical jet up to 20 days before the measurements. Some of the air masses come from the Atlantic lower troposphere north of the ITCZ between the Equator and 10°N more than 20 days before the measurements. Therefore, these air masses have a clean origin, which explains their low $O_3$ content. In contrast in JJASO (Fig. 9.d to 9.f), the air

masses measured at around 25°N come from both east and west. Part of the air masses have been convectively uplifted from South and South East Asian highly polluted boundary layer before being transported over northern Africa at the southern flank of the AMA (Barret et al., 2008). The difference resulting from the uplift of these polluted air masses and from the uplift of cleaner air masses from the Indian Ocean more to the South is highlighted by the IASI CO UT distribution characterized by a clear latitudinal gradient over the Arabian Sea and India (Fig. 5.b). The chemical aging of these polluted

air masses during their transport results in high $O_3$ levels in the North African UT than in DJFM. This important contribution of precursors from Asian anthropogenic sources in JJASO is also highlighted by the SOFT-IO data presented in Fig. 5.k.

It is also interesting to note that the ozone maxima observed beyond the dry zones (by $\approx$10°N in DJFM and by $\approx$15°S in JJASO) are higher in JJASO than in DJFM while CO levels are quite similar (see Fig 4). This probably indicates other sources of precursors not correlated with CO in the southern part. Indeed, previous studies have highlighted a zone of high

$O_3$ mixing ratios over the southern Atlantic during the boreal summer: the Zonal Wave One (Sauvage et al., 2006; Thompson et al., 2003) which results from (i) high precursor emissions in Africa and South America (ii) strong solar radiations, (iii) the high thunderstorm activity and LiNOx production at the ITCZ, (iv) stratospheric $O_3$ incursions and (v) wind currents from the south Atlantic anticyclone.





**Figure 7 : FLEXPART backward plume of air masses where IAGOS flights have recorded maxima (between 6.5 and 11.5°N: a, b and c) and minima (between 13.5 and 8.5°S: d, e and f) of ozone in DJFM. Plumes are represented in term of residence time of particles in each individual output boxes in s, summed on the 20 days backward, on 5° lat of flight and on total column (latitude column for a and d, altitude column for b and e, and longitude column for c and f) and averaged on all studied years. Note that the color scale is different for Longitude-Altitude profiles (a and d), Longitude-Latitude maps (b and e) and Latitude-Altitude profiles (c and f).**

**Figure 8 : FLEXPART backward plume of air masses where IAGOS flights have recorded maxima (between 17.5 and 12.5°S: a, b and c) and minima (between 4.5 and 9.5°N: d, e and f) of ozone in JJASO. Plumes are represented in term of residence time of particles in each individual output boxes in s, summed on the 20 days backward, on 5° lat of flight and on total column (latitude column for a and d, altitude column for b and e, and longitude column for c and f) and averaged on all studied years. Note that the**




color scale is different for Longitude-Altitude profiles (a and d), Longitude-Latitude maps (b and e) and Latitude-Altitude profiles (c and f).



**Figure 9 : FLEXPART backward plume of air masses recorded by IAGOS flights between 20° and 25°N in DJFM (a, b and c) and**
**JJASO (d, e and f). Plumes are represented in term of residence time of particles in each individual output boxes in s, summed on the 20 days backward, on 5° lat of flight and on total column (latitude column for a and d, altitude column for b and e, and**



**longitude column for c and f) and averaged on all studied years. Note that the color scale is different for Longitude-Altitude profiles (a and d), Longitude-Latitude maps (b and e) and Latitude-Altitude profiles (c and f).**

## 7. Conclusions and perspectives

Near-daily IAGOS airborne measurements carried out from end 2005 to 2013 between Europe and Namibia were used to analyze and characterize meridional CO and $O_3$ transects in the African upper troposphere and their seasonal variability. First, we developed a methodology to localize the ITCZ with IAGOS meteorological data (wind and over-water RH data) which allowed us to delimit empirically two periods of the year of strongest interest in this region, namely June-to-October
(JJASO) and December-to-March (DJFM), the intermediate months corresponding to transitional periods. The ITCZ was found to extend over 14.5-4.5°S during JJASO and 4.5-9.5°N during DJFM, in good agreement with NCEP/NCAR reanalysis.

For the two main seasons the CO transects are characterized by peak distributions with maximum mixing ratios located 10° from the position of the ITCZ above the dry regions (132 to 165 ppb at 0-5°N in DJFM and 128 to 149 ppb at 3-7°S in
JJASO). Maximum CO mixing ratios display strong interannual variability with up to 30 ppb between years over the end 2005-2013 period. The $O_3$ transects are characterized by mixing ratio minima of ~ 42-54 ppb at the ITCZ (10-16°S in DJFM and 5-8°N in JJASO) framed by local maxima (~ 53-71 ppb) coincident with the wind shear zones North and South of the ITCZ. The $O_3$ transects display no clear interannual variability in term of gradients and location of the extrema but large mixing ratio variability (~ 15 ppb).

CO and $O_3$ UT African transects are also computed from IASI 2008-2013 retrievals. Comparisons show that IASI is able to capture the main features of the CO and $O_3$ distributions in the African UT. The CO maxima are coincident with IAGOS but the average CO mixing ratios from IASI are biased low by about 20 ppb. Concerning $O_3$, the IASI mixing ratios are in good agreement with IAGOS but the latitudes of the extrema can be shifted by up to 10° relative to IAGOS. The IASI 2D CO and $O_3$ distributions have therefore been very useful to complement the 1D IAGOS meridional transects. IASI UT $O_3$
distributions in DJFM have in particular revealed a crescent-shaped $O_3$ plume above the Atlantic Ocean around the Gulf of Guinea.

SOFT-IO simulations first showed that anthropogenic and fire sources of CO have an impact of the same order of magnitude in the African UT. In DJFM both contributions are coincident and both sources drive the maximum of CO mixing ratio. In JJASO, the observed CO maximum is coincident with the fire contribution. The important Asian anthropogenic contribution
leads to a northward shifted total contribution maximum not coincident with the observed one. This tends to indicate either an overestimation of the Asian anthropogenic contribution or an underestimation of uplifted local contributions to the African UT CO maximum in JJASO.





The anthropogenic CO contribution is mostly local (northern and southern Africa) with a low seasonal variability. CO emitted locally at the surface is advected by the trade winds towards the ITCZ where it is convectively uplifted. The large Asian contribution is related to the fast convective uplift of polluted air masses in the Asian monsoon region which are further westward transported by TEJ and AMA. The fire contribution is almost entirely local with a significant seasonal variability following the alternation of dry and wet seasons.

For the two main seasons, the CO African UT transects characterized by maxima over the dry regions result from the following processes. CO emitted by fires in the dry region is transported towards the ITCZ by the trade winds and further convectively uplifted. In the UT the CO enriched air masses are transported away from the ITCZ by the upper branches of the Hadley cells and accumulates within the zonal wind shear zones where the maxima are located.

The FLEXPART backward plumes revealed that (i) the $O_3$ minima correspond to air masses that were recently uplifted (i.e. for less than 20 days) from the surface where mixing ratios are low at the ITCZ, (ii) the $O_3$ maxima correspond to old high altitude air masses (i.e. uplifted from the surface for more than 20 days before the IAGOS measurement) either locally transported from Africa in the upper branches of the Hadley's cells or transported over long distances from South America by the west winds of the south Atlantic anticyclone and subtropical jets.

(iii) In JJASO, the high $O_3$ concentrations over North Africa around 25°N are related to pollution recently uplifted in the Asian monsoon region and transported south of the AMA.

All these air masses came from areas where $O_3$ precursor emissions are high (Africa, South America and South Asia). We thus do the hypothesis of the $O_3$ formation in these air masses during the transport in the UT which combine high concentrations of precursors (from lightning at ITCZ and surface emissions) and strong radiations to enhance the photochemistry. The more time the air mass spends in the UT and the more $O_3$ is formed (through photochemical production). The comparison of the origin of the air masses recorded by IAGOS flights in the northern hemisphere shows that, in DJFM, without the Asian contribution, the $O_3$ mixing ratios start to decrease. It highlights the importance of the precursor load of air masses to fuel the photochemistry in the UT. In the southern hemisphere, the IAGOS $O_3$ data allow to further characterize the Zonal Wave One phenomenon with similar CO concentrations in DJFM and JJASO but higher $O_3$ concentration in JJASO.

This analysis of meridional transects contribute to a better understanding of distributions of CO and $O_3$ in the intertropical African upper troposphere and the processes which drive these distributions. Therefore, it provides a solid basis for comparison and improvement of models and satellite products in order to get the good $O_3$ for the good reasons.

However, the conclusions of the study which pictures the end 2005-2013 period with IAGOS and could evolve notably according to the projections of future emissions of precursors. Indeed, the analysis of data from the emission inventory MACCity indicates that African anthropogenic emissions of CO have increased by +283% for NHAF and +256% for SHAF between 1960 and 2020 (against +63% for the entire world). CO emissions from these two African regions accounted for 7.87% of global emissions in 1960 compared to 17.96% in 2020. On a smaller time scale (end 2005-2013) this represents a +12% increase in CO emissions for NHAF and +16% for SHAF (+ 1.6% for the entire world). During the same period the



part of both regions in anthropogenic global CO emissions goes from 15% to 16.8%. If African anthropogenic emissions continue to increase as fast as estimated, it could modify the typical distribution of CO and so of $O_3$ with first higher concentrations of CO to fuel $O_3$ formation during all seasons.

**Data availability**

The IAGOS data are available via http://www.iagos.org or directly via the AERIS website (http://www.aeris-data.fr). The SOFT-IO v1.0 products will be made available through the IAGOS central database and are part of the ancillary products (Sauvage et al., 2017c, 2017b, 2017a).

**Competing interests**

The authors declare that they have no conflict of interest.

**Author Contribution**

VL, BS and VT designed the research. All the co-authors contributed to acquisition of data. VL, BS, VT and BB analyzed and interpreted the data. VL drafted the article. VL, BS, VT, HC and BB revised the article.

**Acknowledgments**

The authors acknowledge the strong support of the European Commission, Airbus and the airlines (Lufthansa, Air France, Austrian, Air Namibia, Cathay Pacific, Iberia and China Airlines, so far) who have carried the MOZAIC or IAGOS equipment and performed the maintenance since 1994. In its last 10 years of operation, MOZAIC has been funded by INSU-CNRS (France), Météo-France, Université Paul Sabatier (Toulouse, France) and Jülich Research Center (FZJ, Jülich, Germany). IA- GOS has been additionally funded by the EU projects IAGOS-DS and IAGOS-ERI. The MOZAIC-IAGOS
database is supported by AERIS, the French portal for Data and Service for the Atmosphere (see http://aeris-data.fr for details and list of contributors). The authors would like to thank ECCAD project for providing emission inventories. This project has received funding from the European Union's Horizon 2020 research and innovation programme under the Marie Sklodowska-Curie grant agreement H2020-MSCA-COFUND-2016-754433.

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
