# Peer review of "Origins and characterization of CO and O3 in the African upper troposphere"

_Atmospheric Chemistry and Physics, 2021_

## Author Response (AR1)

**Author Responses to Referees and comments on**

**Origins and characterization of CO and O3 in the African upper troposphere**

Victor Lannuque1,a, Bastien Sauvage1, Brice Barret1, Hannah Clark4, Gilles Athier1, Damien Boulanger2, Jean-Pierre Cammas5, Jean-Marc Cousin1, Alain Fontaine1, Eric Le Flochmoën1, Philippe Nédélec1, Hervé Petetin6, Isabelle Pfaffenzeller2, Susanne Rohs3, Herman G.J. Smit3, Pawel Wolff1 and Valérie Thouret1

1Laboratoire d'Aérologie, CNRS, UPS, Toulouse, France 2Observatoire Midi-Pyrénées, Université de Toulouse, CNRS, UPS, Toulouse, France 3Forschungszentrum Jülich GmbH, Institut für Energie- und Klimaforschung, IEK-8 Troposphere, 52425 Jülich, Germany 4IAGOS-AISBL, 98 Rue du Trône, Brussels, 1050, Belgium 5Observatoire des Sciences de L'univers de la Réunion, UMS3365, la Réunion, France 6Barcelona Supercomputing Center, Barcelona, Spain aNow at: CEREA, École des Ponts, EDF R&D, 77455 Marne-la-Vallée, France Lannuque (victor.lannuque@enpc.fr) Bastien Sauvage Correspondence to: Victor and (bastien.sauvage@aero.obs-mip.fr)

We thank the reviewers for their comments and suggestions on the manuscript. We outline below responses to the points raised by each referee and summarize the changes made to the revised manuscript. We have also provided a revised version of the manuscript with changes appearing highlighted in yellow in the text.

**Reponses to RC1**

> Generally speaking, titles don't have a period at the end. Please correct.

Titles were modified

> The abstract is too long and has redundant information. For example, L31-34 is basically repeated in L37-39. I would suggest to rework the abstract to make it more concise.

The abstract was modified to be shorter and redundant informations were removed.

> P9 L254 I would refer to easterly winds rather than to 'Zonal winds < 0' in the second sentence of this line. I would also add units (m/s) to the 0.

The text was modified.

> P10 L264 The transition period is sometimes written as April-May and sometimes as AM (e.g. L265). I would suggest to use only the abbreviation after the definition in L246.

AM (and N for November) are not used in the text. We have decided not to use the abbreviations for the April-May and November transition periods in the text to avoid misunderstandings (with AM from AM-

PM and N for North which are more commonly used in literature). The abbreviations are however used in the figures and are therefore now presented in the text with the mention: "in the tables and figures".

> Fig. 2 presents relative humidity as a fraction of 1, while other plots shows it as percentage. Please correct to improve consistency. I would also remove the ticks on the y axis of panels i,j,k and l, as they are not coincident with the boxes shown. It might also be a good idea to show the NCEP output as a grey shading in the same way IASI is introduced in Fig. 4. That would provide a more quantitative comparison between the IAGOS in-situ met data and NCEP.

As recommended, figure 2 was modified: relative humidity is now presented in percentage, y-axis ticks was removed for panels i, j, k and l, and the NCEP data range are now represented by grey areas.

> P22 L508 – LiNOx is lightning NOx? Please clarify (e.g. add abbreviation description)

The abbreviation is now presented where the lightning  $NO_X$  are first clearly mentioned in the text in section 6.

> P22 L487 '(which may be relevant????? should maybe mention "this)' looks like a comment added by the authors during writing process.

You are right. The comment was removed.

> Fig. 5. Please make the labels larger (specially the lat, lon ones).

Figure 5 was modified as requested.

> Figs. 7, 8 and 9 show several levels in the troposphere and just one between 11 and 50 km. Does this level intend to show the stratospheric contribution (i.e. stratospheric intrusions)? Does it correspond to the average of several levels? 11 km doesn't seem to correspond to the stratosphere in the tropical region. Please clarify.

Referee makes a good point, as this part was not clear enough. We do not intend to show stratospheric contribution here because it would request a deep investigation to determine the tropopause altitude, which could be complicated at the tropics using for instance potential vorticity. The goal is just to determine upper tropospheric transport. To go a little further in the analysis, we see that the contributions of the "11km and above" layer does not extend beyond the tropics (+/- 25 ° approximately). This means that we mainly have contributions from the UT because the Stratosphere to Troposphere exchanges are negligible in the tropics, or there is on the contrary troposphere to stratosphere transport. Text and figures 7, 8 and 9 were modified to mention that and to consider only altitude higher than 11km.

> P28 L617 Join (iii) to the previous paragraph.

The text was modified.

> P33 L846-849 "Thouret, V., Marenco, A., Sabatier, P., Logan, J. A., Ndec, P. and Grouhel, C.: Comparisons of ozone measurements from the MOZAIC airborne program and the ozone sounding network at eight locations Goose is obviously one important to its has followed by a recent impact and Marerico greenhouse Staehelin Copyfight by the American Geophy, J. Geophys. Res., 103, 1998." -> (?) Please revise this citation.

The citation was corrected.

> I would suggest the authors to carefully proof-read the document and look for consistency issues. For example, figures are referred as Fig., fig, Fig, etc. without consistency.

The text was proof red to correct these inconsistencies and to follow the Copernicus guidelines (only "Figure" and "Fig." are conserved).

**Reponses to RC2**

> 1) Period data coverage: The data used in this paper stop in 2013. Did the measurements stop in 2013, or were they continued but not used for this work? In the first case, it must be clearly indicated in the text. In the second case, why more recent data were not used?

As mentioned in the introduction, the almost daily IAGOS flights between Namibia and Europe only took place between 2006 and 2013. The precision "equipped with IAGOS instruments" has been added for clarity.

> 2) Life-time of CO/O3 vs SOFT-IO time backtrajectory: The Soft-IO time backtrajectory is 20 days (line 144). The lifetime of CO in the troposphere is larger than 20 days, then the analysis takes into account only recent contributions of CO. Can the authors assess whether this approximation significantly influences the results or not?

You are right. Average CO lifetime is about 40 days in troposphere. The purpose of SOFT-IO is to give an indication of the origins of the CO anomalies, i.e. the CO recently emitted by anthropogenic or fire sources. Considering back trajectories for more than 20 days would cause several problems: (i) an increase in computing time, (ii) an increase in uncertainties on trajectories and (iii) a more difficult estimation of the CO quantity because its potential reactivity during its transport is not taken into account (uncertainty limited by using shorter durations). Taking 20 days as the duration for the back trajectories, we already see masses of air coming from the other side of the globe. Moreover test has been realized during SOFT-IO development and back-trajectories longer than 20 days have a little influence in the CO calculations (either intensity or origin) with increased dispersion over time

> 3) differences between IASI and IAGOS: Figure 5 presents seasonal IASI maps of CO and O3, with IAGOS points superimposed. This highlights that IAGOS CO and O3 values are systematically largely over IASI values, for the two gazes and the two seasons. The authors mention that these discrepencies and biases are due to very different natures of observations (line 391). Why not but I think that this point is important and the text has to be clear and precise. IAGOS overestimates O3 and CO or IASI underestimates ? Why exactly ? I have read the two following papers: "Maya George, Cathy Clerbaux, Idir Bouarar, Pierre-François Coheur, Merritt N. Deeter, et al., An examination of the long-term CO records from MOPITT and IASI: comparison of retrieval methodology, Atmospheric Measurement Techniques, European Geosciences Union, 2015, 8 (10), pp.4313-4328" and "Safieddine, S., Boynard, A., Hao, N., Huang, F., Wang, L., Ji, D., Barret, B., Ghude, S. D., Coheur, P.-F., Hurtmans, D., and Clerbaux, C.: Tropospheric ozone variability during the East Asian summer monsoon as observed by satellite (IASI), aircraft (MOZAIC) and ground stations, Atmos. Chem. Phys., 16, 10489–10500, https://doi.org/10.5194/acp-16-10489-2016, 2016", But I could not find evident reason for the discrepencies of the present study. Could it be a problem of difference of altitude of observation between the two ?

The two references cited by the reviewer are not really appropriate for comparisons with the results presented in our paper. George et al. compare integrated columns from two satellite nadir thermal infrared sounders (IASI and MOPITT) with similar sensitivities and coarse vertical resolutions. Therefore they cannot detect biases related to the remote nature of the satellite observations. In Safieddine et al. there are comparisons for the 0-6 km integrated columns between IAGOS and IASI-FORLI data for  $O_3$  and CO at Asian airports. Because of differences in altitude range, but also product and region, it is not appropriate for our study concerning SOFRID O3 and CO in the UT over Africa. SOFRID CO has been validated over Africa by De Wachter et al. (2012) which is cited in our paper. Nevertheless, in De Wachter et al., we make quantitative comparisons with partial columns from IAGOS

vertical profiles at take-off and landing. Here, as stated in the manuscript, we make direct comparisons with IAGOS cruise data in the UT. The results are therefore not comparable to De Wachter et al. (2012). The aim here is to evaluate the features observed by IASI in the UT in order to use IASI data to characterize the CO and  $O_3$  2D UT distributions to complement IAGOS latitudinal African transects. As IAGOS provides in-situ observations at a precise altitude and IASI remote sensing data with a broad vertical resolution (~ 6km) the comparison is only qualitative.

Concerning  $O_3$ , the largest discrepancies appear between ~5°N and ~25°N with a large undersestimation of SOFRID versus IAGOS. These discrepancies are probably resulting from the bad representation of the emissivity of arid and desert surfaces which impacts the IASI radiances in the  $O_3$  band used for  $O_3$  retrievals.

In order to provide some clues about SOFRID versus IAGOS discrepancies we have added the following statements:

L~390 for CO: "Indeed, with a coarse resolution of about 6 km (De Wachter et al., 2012) SOFRID-CO at 253 hPa is a weighted average of the profile over +/- 3km around this level. The most significant underestimation of SOFRID-CO occurs around the latitudinal maxima (Fig. 4) which is caused by the advection of the convective outflow loaded with fire products. SOFRID at 253 hPa therefore results from the smoothing of the maximum with lower values from above and below 253 hPa."

L~378 for O3: "The extrema discrepancies between IASI and IAGOS in DJFM and JJASO mostly result from an underestimation of IASI O3 relative to IAGOS between 5-10°N and the northern boundary of the tropics at ~25-30°N. This underestimation is most probably related to the poor representation of the surface emissivity by the climatology used by RTTOV over arid and desert surfaces in the 10  $\mu$ m region. Such a retrieval problem over desert regions has been documented by Boynard et al. (2018) for FORLI-O3."

**> 4) Minor point: Line 386: The text mention four seasons but the analysis is based on a 2 seasons separation**

There are indeed 4 seasons or periods that we have divided into two groups: two main seasons (DJFM and JJASO) on which the study mainly focuses and two transition periods (AM and N) which are also studied but less in detail. Maybe the misunderstanding comes from an amalgamation between "seasons" (which are 4) and "main seasons" (which are 2).